# Synthesis and Characterization of Polyimides with Naphthalene Ring Structure Introduced in the Main Chain

**DOI:** 10.3390/ma15228014

**Published:** 2022-11-14

**Authors:** Jiang-Rong Luo, Yi-Dong Liu, Heng Liu, Wei-Peng Chen, Ting-Ting Cui, Liangang Xiao, Yonggang Min

**Affiliations:** School of Materials and Energy, Guangdong University of Technology, Guangzhou 510006, China

**Keywords:** naphthalene-containing cyclic diamine, terpolymer, polyimide film, dielectric properties

## Abstract

In this paper, a new aromatic diamine monomer 4,4′-(2,6-naphthalenediyl)bis[benzenamine]) (NADA) was synthesized and a series of modified PI films containing naphthalene ring structure obtained by controlling the molar ratio of NADA monomer, ternary polymerization with 4,4′-oxydianiline (ODA), and pyromellitic dianhydride (PMDA). The effects of the introduction of the naphthalene ring on the free volume and various properties of PI were investigated by molecular dynamic simulations. The results show that the comprehensive properties of the modified films are all improved to some extent, with 5% thermal weight loss temperature (T_d5%_) of 569 °C, glass transition temperature (Tg) of 381 °C, tensile strength of 96.41 MPa, and modulus of elasticity of 2.45 GPa. Dielectric property test results show that the dielectric constant (Dk) of the film at 1 MHz is reduced from 3.21 to 2.82 and dielectric loss (Df) reduced from 0.0091 to 0.0065. It is noteworthy that the PI-1 dielectric constant is reduced from 3.26 to 3.01 at 10 GHz with only 5% NADA doping, which is expected to yield the best ratio and provide the possibility of industrial production.

## 1. Introduction

With the rapid development of the microelectronics industry, especially the emerging fifth-generation mobile communication technology (5G technology), the number of transistors integrated on the chip in integrated circuits is increasing dramatically, and the density of interconnected wiring in the chip is also increasing dramatically. Therefore, the parasitic capacitance effect generated by resistance and capacitance (RC) delay is becoming more and more serious, leading to problems such as excessive power consumption and signal delay caused by capacitance and reactance [1,2,3,4,5]. The dielectric constant (Dk) of communication materials will directly affect the speed of signal transmission, and the high dielectric loss (Df) will lead to signal distortion. In order to meet the needs of high speed and accuracy of information transmission in the era of 5G communication, as well as light weight of communication equipment, and to provide technical support for applications such as autopilot, telemedicine, smart cities, and finally realizing the Internet of Everything, it is urgent for us to develop and prepare dielectric materials with better insulation and lower dielectric constant [6,7].

Polyimide is considered an ideal material for next-generation high-performance interlayer dielectrics in microelectronic devices because of its excellent properties, such as lower dielectric constant, good thermal stability and chemical stability; however, the dielectric constant of conventional PI films is around 3.49 (≤1 MHz frequency) [8], which still cannot meet the requirements of the microelectronics industry for the continuous reduction of dielectric material Dk [7,9,10,11,12,13]. In addition, the dimensional change of polyimide films during long-term use may lead to accidents such as disconnection and short circuits in the circuit structure of thinner liners [14]. Therefore, we hope to further reduce the dielectric constant of PI film by means of modification and improve its dimensional stability, elastic modulus, and moisture absorption performance.

Common modification methods to reduce the dielectric constant of polyimide are to reduce the molecular polarization rate of the material and to introduce nanoscale pores inside the polyimide [15,16]. The former is mainly to introduce fluorine atoms or fluorine-containing groups in the molecular chain, which are highly electronegative and can effectively fix electrons, thus reducing the polarization rate of the polymer [17,18]. Min Zhong et al. [19] synthesized a series of polyimides containing m-trifluoromethylphenyl, in which the dielectric constants (Dk) of 2-6FDA films were less than 2.66 and 2.38 at 1 MHz and 10 GHz frequencies, respectively. The latter was achieved by introducing nanoscale pores, thus reducing the content of polarized molecules per unit volume—in short, more pores can lead the lower dielectric constant. Zian He et al. [20] prepared polyhedral oligomeric silsesquioxanes (POSS) contained in a hierarchical porous structure, which was prepared by the self-assembly of water droplets, used as a coating for flat polyimide (PI) film with a low PI dielectric constant of 2.42 at a frequency of 1 MHz.

With the increase in elemental fluorine content as well as porosity, it often brings problems such as high raw material cost, generation of corrosive by-products of hydrogen fluoride and fluorine gas, poor adhesion affecting subsequent processing, and inevitable collapse of pores leading to degradation of mechanical properties [21,22]. Therefore, there is an urgent need for a more economical method to prepare low dielectric materials with good heat resistance but without sacrificing mechanical properties. In this paper, a diphenylamine monomer with a naphthalene ring was synthesized by Suzuki reaction, and a series of PI films containing a naphthalene ring structure was prepared by changing the molar ratio of NADA monomer based on the existing PMDA/ODA diamine and dianhydride monomer system for ternary polymerization. The tested results showed that with the addition of NADA monomer, the performance of PI films improved, with Dk decreasing from 3.21 to 2.82 and Df decreasing from 0.0091 to 0.0065 at 1 MHz. Notably, Dk decreased to 3.01 (10 GHz) when the NADA molar ratio was 5%.

## 2. Experimental

### 2.1. Materials

2,6-Dibromonaphthalene (99%): Macklin; 4-aminobenzeneboronic acid pinacol ester (99%): Aladdin; tetrakis(triphenyl (phosphine)palladium [Pd(PPh_3_)_4_, 98%]: Macklin; tetrabutylammonium bromide (TABA, 99%):Aladdin; 1,4-dioxane (99%):Aladdin; (K_2_CO_3_, 99%):Aladdin; pyromellitic dianhydride (PMDA, 99%); 4,4′-oxydianiline (ODA, 98%); N,N-dimethylacetamide (DMAc, 99.8%).

### 2.2. Monomer Synthesis

The synthetic route of the diamine monomer 4,4′-(2,6-naphthalenediyl)bis[benzenamine] (NADA) is shown in Figure 1.

Under the protection of nitrogen, 2,6-dibromonaphthalene (2.86 g, 0.01 mol), 4-aminophenylboronic acid pinacol ester (6.56 g, 0.03 mol), tetrabutylammonium bromide (TBAB, 1.00 g, 10 wt%), catalyst tetrakis(triphenylphosphine)palladium [Pd(PPh_3_)_4_, 0.20 g, 2 wt%] were added to a 500 mL three-necked flask. After stirring for 10 min, add 60 mL of K_2_CO_3_ (2 mol/L), and stirring at 75–85 °C for 36 h. After the reaction, a deep green mixed solution is obtained, using 100–200 mesh neutral alumina as stationary phase and petroleum ether/ethyl acetate (2:3, *v*/*v*) as mobile phase, the solvent is removed after rotary evaporation, and the diamine monomer NADA yellow powder was obtained after vacuum drying 2.18 g, yield 70.32% and 95% purity. As shown in Figure 1, ^1^H NMR (400 MHz, DMSO-*d*6) δ 8.01 (d, *J* = 1.8 Hz, 2H), 7.91 (d, *J* = 8.6 Hz, 2H), 7.74 (dd, *J* = 8.5, 1.7 Hz, 2H), 7.57–7.49 (m, 4H), 6.73–6.66 (m, 4H), 5.30 (s, 4H).MS(*m*/*z*):311.15([M+H]^+^).

### 2.3. Preparation of Polyimide Films

Figure 2 describes the preparation of a new terpolymer PI film, which mainly consists of a two-step synthesis and thermal imidization process. The mole ratio of different monomers is shown in Table 1, labeled as PI-1–PI-4 and control PI-0. PI-4 was used as an example to illustrate the preparation process of PI. The diamine monomer NADA (0.62 g, 0.002 mol) and ODA (1.60 g, 0.008 mol) were added to a 50 mL beaker, and DMAc (17.60 g) was added and stirred until complete dissolution, then PMDA (2.18 g, 0.01 mol) was slowly added, and this solution was stirred at room temperature for 12 h. After vacuum evacuation, a solid content of 20% of polyamide acid (PAA) solution was obtained after vacuum defoaming. The PAA solution was uniformly spin-coated onto a clean glass plate by a gel machine, and then placed in an oven at 90 °C for 30 min for removing the solvent to obtain the cured PAA film. The PAA film was uncovered by a blade and fixed by tensioning with a special metal frame and clips, and finally, it was placed in the oven at 250 °C for 10 min and 350 °C for 10 min to complete the imidization, and the thickness of the obtained PI films was controlled at 30–50 μm.

### 2.4. Testing and Characterization

Gel permeation chromatography (GPC): measured by a U3000 high performance liquid chromatograph from Thermo Fish, Shanghai, China.

Dynamic viscosity test: NDJ-1S digital viscometer of Shanghai Hengping Company of China was used for testing.

Liquid phase mass spectrometry: The TSQ Endura ultra-high performance liquid chromatography tandem triple quadrupole mass spectrometer from Thermo Fisher Scientific, Shanghai, China was used for determination.

Nuclear magnetic resonance (^1^H-NMR) testing: AVANCEIIIHD400 NMR instrument from Bruker, Switzerland was used for the determination.

Infrared spectroscopy (FT-IR) testing: Scanning was performed using a Nicolet ATR-60 Fourier transform infrared spectrometer from Thermo Fish, Shanghai, China.

Thermogravimetric analysis (TGA): The TGA/DSC3+ thermogravimetric analyzer from Mettler Toledo, Zurich, Switzerland was used for the test. The temperature was increased to 800 °C at a flow rate of 50 mL/min in a nitrogen atmosphere at 10 °C/min.

XRD testing: Bruker D8ADVANCE X-ray diffractometer, Switzerland, Cu target, tube voltage 40 kV, tube current 40 mA, scanning angle 2θ range 6° to 60°, scanning speed 6°/min.

Mechanical property testing: The AG-X plus electronic universal material testing machine from Shimadzu, Japan was used for testing. The sample size was a rectangular film strip of 1 × 5 cm, the stretching rate was 50 mm/min, and the test was repeated three times.

Thermomechanical analysis (TMA): The test was performed with a TMA-Q400EM thermomechanical analyzer from TA Instruments, New Castle, USA, with a rectangular sample strip of 5 mm width and a constant tensile force of 50 mN.

Contact angle: Using SL200KB contact angle tester from KINO Industrial, Boston, MA, USA.

Water absorption: The PI film was cut into square pieces of 3 cm × 3 cm, dried in an oven at 60 degrees for 24 h, and weighed for mass m_1_. Then it was soaked in deionized water for 12 h, and weighed immediately after wiping the water from the film surface to obtain mass m_2_. Each sample was measured three times and the average value was taken. Calculated from the formula (m_2_ − m_1_)/m_1_ × 100%.

Dielectric constant and loss: The impedance analyzer WY2818A of Shanghai Wu Yi Electronic Equipment Co., Ltd. of China was used to determine at 1 MHz frequency.

### 2.5. Computational Details

This experiment mainly investigates the dielectric properties of modified PI, Dk and Df decrease with the increase of Fractional Free Volume (FFV), so the simulation calculation is carried out for its FFV and provides some theoretical support for the PI performance. FFV were performed for PI using Materials Studio (MS) 2019 software. First, repeat structural units of PMDA/ODA/NADA were drawn, the degree of polymerization was set to 10, the ratio of naphthalene ring structure was controlled to construct PI-0–PI-4 molecular chains, and geometry optimization was completed. Then, in order to further reduce the impacts of edge effect and chain end effect, the polyimide periodic structure was constructed, and the four molecular chains were modeled in the periodic structure and their structures were optimized to obtain a stable structure. The isothermal isovolumetric thermodynamic conditions (NVT) were annealed from 300-700-300 K for 5 cycles of temperature increase and decrease to eliminate the unreasonable structures generated during the modeling process. Through molecular dynamic simulation of the NVT system, the difference between the upper and lower temperature fluctuations of the simulated polyimide system is less than 10% and the energy fluctuation range does not exceed 3% for a fixed time of 500 ps at 300 K, which proves that the structure is completely relaxed and in equilibrium.

## 3. Results and Discussion

### 3.1. Polyamide Acid Performance Analysis

The relative molecular weight and dynamic viscosity of polyamide acids (PAA) were characterized by gel permeation chromatography (GPC) and viscometer, respectively. As shown in Table 1, the relative molecular weight of PAA showed a significant decrease as the molar amount of NADA monomer increased, while the dynamic viscosity increased from 24.4 Pa.s to 28.14 Pa.s. This was due to the introduction of the naphthalene ring structure, which strengthened the intermolecular forces and made the relative movement of PAA molecular chains more difficult, and thus the viscosity increased slightly. The low reactivity of NADA compared to ODA makes the overall polymerization degree decrease, and secondly, the rapid increase of viscosity will further hinder the chain growth and make the relative molecular weight of PAA decrease.

### 3.2. Fourier Transform Infrared Spectroscopy Analysis

The chemical structure of PI was characterized using FT-IR spectroscopy, as shown in Figure 2. The five groups of PI films after thermal imidization showed obvious characteristic absorption peaks at 1778, 1720, 1370 and 725 cm^−1^, specifically near 1778 cm^−1^ (imide C=O asymmetric pair stretching vibration) and 1720 (imide C=O asymmetric pair stretching vibration) for the imide ring, and near 1370 cm^−1^ (stretching vibration of imide ring C-N), and near 725 cm^−1^ (bending vibration peak of imide C=O). In addition, no carboxyl group (-COOH), amide (-CO-NH-) or residual amino group (-NH-) appeared in the range of 3300–3500 cm^−1^. The characteristic absorption peaks indicate that the polyimide has been completely imidization.

### 3.3. XRD Date

In order to study the internal aggregation structure of PI molecules, the morphology of PI films was analyzed by XRD. As shown in the Figure 3, PI-0–PI-4 exhibited more obvious diffraction peaks at 2θ of 18°–21°, and the diffraction peaks gradually shifted to higher angles and the diffraction intensity gradually flattened. In addition, we calculated the diffraction angle corresponding to each polyimide by XRD, and the relationship between diffraction angle and d-spacing can be established by the Bragg equation (2dsinθ = nλ), and the calculation results are shown in Table 2. In general, d-spacing reflects the average distance between polymer chain segments, and indirectly reflects the free volume in the compound system. This is due to the fact that there are many structural units in a polymer chain, so the interaction of the structural units is important for its aggregation state. With the decrease in flexible ether bonds in ODA and the increase of rigid naphthalene rings in NADA, the inward rotation of the molecular chains decreases and tends to be rigid, the π–π overlap of benzene and naphthalene rings in turn makes the molecular chains tend to be densely stacked, which eventually leads to the ordered arrangement of the polymer. It is noteworthy that the chain spacing of PI-1 is slightly increased compared with PI-0, which is likely due to the small amount of naphthalene ring structure that makes the molecular chains less regular, thus leading to the increase in free volume of the system.

### 3.4. FFV Analysis

This experiment mainly investigates the dielectric properties of modified PI. Dk and Df decrease with increasing fractional free volume (FFV), so simulation calculations are performed for its FFV and provide partial theoretical support for PI performance. The FFV of the PMDA/ODA/NADA system polyimide at 300 K was calculated with the Atom Volume and Surface toolbar of MS, where the filled gray area in Figure 4 indicates the volume occupied by the polymer, and the filled blue area indicates the free volume. The total volume of PI molecular cells and the free volume of the system are calculated, and the free volume divided by the total volume to the free volume fraction of the system. As shown in Table 3, the introduction of the naphthalene ring structure makes the FFV of PI-1 decrease significantly, indicating that the biphenyl structure strengthens the intermolecular forces and makes the chain spacing narrower. The increase in FFV of PI-1 to PI-2 may be due to a certain amount of non-planar conjugated naphthalene ring structure, which plays a certain steric hindrance effect, and then influences the dense stacking of molecular chains. After which, as the naphthalene ring increases, the π–π stacking plays a dominant role in making the FFV decrease continuously.

### 3.5. Polyimide Thermal Properties and Coefficient of Thermal Expansion

The thermal stability of the polymer films was studied by thermogravimetric analysis (TGA), and the TGA curves of PI films are shown in Figure 5. From Table 4, it can be seen that the thermal decomposition temperature (T_d5%_) of PI-0–PI-4 at 5% weight loss is in the range of 553–569 °C, the thermal decomposition temperature (T_d10%_) at 10% weight loss is in the range of 573–586 °C, and the residual mass fraction (R_w800_) of the film at 800 °C is in the range of 50.01–60.06%. Its thermal stability also increases gradually with the increasing of the molar content of diamine monomer NADA, and the variation of T_d5%_ is larger than that of T_d10%_. In addition, with the increasing of the content of aromatic ring in the PI molecular chain, R_w800_ also appears to be significantly improved. Since NADA is a diamine monomer with a rigid structure, when its molar content increases it also increases the overall rigidity of its molecular chain, and the energy required to break the C-C bond in the biphenyl structure is higher than the energy required to break the ether bond in ODA, so the thermal decomposition temperature T_d_ increases to some extent. In addition, we also found that the improvement of thermal stability of PI is more obvious with the amount of NADA around 5%, and then the improvement of thermal stability slows down significantly with the increase of NADA content.

Numerous structural studies have shown that the coefficient of thermal expansion (CTE) tends to decrease with increasing in-plane orientation, and the degree of rigidity of the PI backbone will directly affect the in-plane orientation [23]. Therefore, the PMDA/ODA/NADA system PI was prepared by introducing a rigid naphthalene ring structure through copolymerization method to further reduce its CTE value. The dimensional stability of the films was determined by the TMA method and evaluated by the coefficient of thermal expansion (CTE). As shown in Figure 6, plotting the corresponding dimensional change–temperature curve of PI film, the CTE values were calculated from the slope of the curve from 100 °C to 200 °C (before T_g_). As shown in Table 4, the CTE of PI films decreased from 50.01 ppm/K to 41.13 ppm/K as the molar content of NADA monomer increased, which may be due to the rigid structure of the PI backbone that makes it easier for the spontaneous in-plane orientation of its molecular chains, resulting in a reduction in molecular chain spacing and more of a tendency for dense stacking, which eventually caused the lower CTE values of PI films. This is in agreement with the above XRD test analysis.

The glass transition temperature of this group of PI films was analyzed by TMA test, and the glass transition temperature T_g_ was determined by the intersection of the tangent lines of the curves before and after the glass transition, as shown in Table 4, with the increase of the molar content of the diamine monomer NADA, T_g_, increasing from 350 °C to 381 °C. XRD proved to lead to the reduction of the proportion of single bonds on the chain that can be internally rotated with the introduction of the rigid naphthalene ring structure, and the reduction in rotatable space of the molecular chain, which in turn elevated the glass transition temperature of PI films.

### 3.6. Mechanical Properties of PI Films

To investigate the effect of NADA on the mechanical properties of PI films, the tensile properties (tensile strength, elongation at break and modulus of elasticity) of PI films were measured, as shown in Table 5. The introduction of the naphthalene ring structure improved the regularity of the chain segments, which enhanced the mechanical properties of PI films to some extent, with tensile strength, elastic modulus and elongation at break ranging from 81 to 96 MPa, 2.14 to 2.47 GPa and 21.16 to 16.51%, respectively. From the molecular structure analysis, the rigidity and motility of the monomer determine the physical properties of the polymer to some extent. In the PMDA/ODA/NADA system, as the molar ratio of the two diamine monomers changes, the structure and properties of the molecular chains in the PI films change accordingly. The presence of flexible ether bonds allows the PI films to consume energy better through the stretching and structural deformation of molecular chains when subjected to the same magnitude of strain, thus showing lower elastic modulus and higher elongation. In contrast, the introduction of NADA monomers results in narrower chain spacing and restricted creep space of molecular chains, while the rigid biphenyl structure makes the molecular chains more prone to brittle fracture [24]. Therefore, the mechanical property test results show that the introduction of NADA monomer can significantly improve the tensile strength and elastic modulus of PI films of this system, while the elongation at break tends to decrease. It is not difficult to find that when NADA monomer is incorporated into only 5%, it has a certain ability to improve the comprehensive mechanical properties of PI. The introduction of a small amount of rigid naphthalene ring structure is proved by XRD, which slightly increases the free volume of the polymer and gives more space for chain segments to rotate, and at the same time, brings local support points to the relatively flexible chain segments of PMDA/ODA system, which plays a kind of “rigid-flexible” effect.

### 3.7. Water Absorption and Dielectric Properties of PI Films

Hydrophobicity is critical to the reliability of devices in use, especially in microelectronics. Therefore, water contact angle and water absorption tests were used, and surface hydrophobicity was characterized by the contact angle of water droplets on PI films surface, which is shown in Figure 7. As the content of NADA monomer increases, the water contact angle of PI-0 increases from 66° to 78° for PI-2. It is noteworthy that when the content of NADA monomer reaches 15%, the contact angle of PI-3 decreases to 70° instead, and the contact angle size of the subsequent PI-4 does not change significantly, but is still larger than that of PI-0. In addition, the water absorption test in Table 6 also indirectly proves that PI-1–PI-4 have good hydrophobicity, and their overall water absorption decreases between 0.17% and 0.34% compared to the control PI-0, where PI-2 shows a relatively lower water absorption of 0.91%. This is due to the fact that the ether bond in ODA acts as a donor of electrons and forms stable hydrogen bonds with hydrogen in water [25]. With the reduction of the hydrophilic group ether bond and the introduction of the non-polar group naphthalene ring effectively improves the surface hydrophobicity of PMDA/ODA membranes.

PI is widely used as an electrical insulating material in the electronics industry with high requirements for dielectric properties. The dielectric properties of all films at 1 MHz are compared in Table 6 and it was found that the dielectric constant and dielectric loss of PI films both decreased significantly as the content of NADA increased. Thus, the optimum dielectric constant reaches 2.82 (PI-4 at 1 MHz) and the dielectric loss remains at 0.0065. In general, the reduction of dielectric constant and dielectric loss is usually achieved by the design of the polymer molecular structure. The introduction of low polarizability substituent groups in the polyimide chain, thus reducing the material polarizability or the introduction of asymmetric monomers or large side groups in the molecule to increase the free volume of the PI molecule. However, the decrease in the dielectric constant of this series of PIs is unusual, both in terms of the higher polarization rate due to the introduction of the conjugated structure naphthalene ring, or in terms of the XRD results showing a narrower chain spacing resulting in a lower FFV. Notably, comparing the dielectric properties of all films at 10 GHz, the dielectric constants and dielectric losses of NADA-doped monomer PI are lower than those of pure PI-0, of which PI-1 shows a lower dielectric constant of 3.01. Based on the XRD characterization and FFV simulation results, it is speculated that it is probably because a small amount of naphthalene ring structure plays the role of non-coplanar large planes, which have a certain bit resistance effect. This in turn influences the dense stacking of molecular chains, which leads to an increase in the free volume of the system.

## 4. Conclusions

In summary, a new diamine monomer (NADA) with a naphthalene ring-containing structure was successfully prepared by Suzuki reaction. By introducing the rigid NADA monomer into the PI of PMDA/ODA system, the intermolecular force was enhanced, and the molecular chains tended to be tightly stacking with narrower and more regular spacing, which effectively reduced the CTE and improved the thermal and mechanical properties. The comparative performance analysis shows that at 1 MHz, the PI-0–PI-4 dielectric constants (Dk) and dielectric losses (Df) show a continuous decreasing trend with increasing molar ratio of NADA monomers, decreasing to 2.82 and 0.0065, respectively. It is probably because a small amount of non-coplanar naphthalene ring structure decreases the molecular chain regularity, which leads to an increase in the free volume of the system, resulting a lower dielectric constant of PI-1 at 10 GHz frequency, which is expected to be achieved by the molecular structure design through the regulation of the diamine monomer ratio, and the targeted enhancement of a certain aspect of performance through the selection of suitable components and ratios to broaden the application avenues while reducing the raw material cost.

## Data Availability

Not applicable.

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
