# Peer review of "Synthesis and Characterization of Polyimides with Naphthalene Ring Structure Introduced in the Main Chain"

_materials, 2022, doi:10.3390/ma15228014_

Round 1

Reviewer 1 Report

I recommend:

1) To enlarge the figure summarizing IR spectra.

2) To add the molecular parameters of the polyamic acids (e.g. based on  viscometric data).

3) To discuss in more detail the final composition of the polymer products (i.e. theory vs experiment).

4) To align the numbering of the tables and figures with the text.

5) To adjust the accuracy of some results (e.g. elongation xx.xx %).

Author Response

Thank you very much for your time and consideration! We are grateful for the constructive and insightful comments from the Reviewers and have revised the manuscript accordingly. Each of the Reviewers’ comments and suggestions is addressed point-by-point below. A marked version of the manuscript is provided. We believe that the revised manuscript is greatly improved and meets all the requirements for publication in Materials. I hope you find our responses and revisions satisfying and thank you for your kind contributions.

Sincerely yours

Liangang Xiao, Professor
School of Materials and Energy
Guangdong University of Technology
Guangzhou, 510006, China
Email: xiaolg@gdut.edu.cn  

  • To enlarge the figure summarizing IR spectra

Response: Enlarged some of the images (Figure 2, Figure 3, Figure 5 and Figure 6) to make the data clearer.

  • To add the molecular parameters of the polyamic acids(e.g. based on viscometric data)

Response: The reviewer's suggestion on the addition of molecular weight and viscosity data for PAA is greatly appreciated. It avoids misunderstanding caused by the polymerization process and makes the performance data more credible. Added the relative molecular weight and dynamic viscosity parameters of polyamide acids in Table 1, and added the model numbers of gel permeation chromatography and viscometer in heading 2.4. In addition, the analysis of polyamide acid was added to heading 3.1, increasing the number of headings in the results and analysis to 3.7.

  • To discuss in more detail the final composition of the polymer products(i.e. theory vs experiment).

Response: The performance of the monomeric NADA part was added at heading 2.3 and why PMDA/ODA/NADA terpolymerization was used to enhance the rationality of the experimental design and also to make the line logic.

  • To align the numbering ofthe tables and figures with the text

Response: Changed the insertion format of Figures 4 and 7 so that the images, tables, and corresponding text are aligned to the left.

5)To adjust the accuracy of some results(e.g. elongation xx.xx%).

Response: The mass parameters of the monomer weighing during the preparation of PI in heading 2.3 are accurate to two decimal places (uniform accuracy).

Reviewer 2 Report

The preparation and characterization of a diamine monomer with a naphthalene ring structure added to the main chain were explored by the authors.

From an academic perspective, this study seems to be significant for the field of material science.

Minor changes are suggested. Please read the questions and leave a comment below.

- The dielectric constant values are far too precise. The values, in my opinion, should be revised to ensure maximum one decimal place precision.

- The collected results should be compared to what is known from the literature.

Author Response

Thank you very much for your time and consideration! We are grateful for the constructive and insightful comments from the Reviewers and have revised the manuscript accordingly. Each of the Reviewers’ comments and suggestions is addressed point-by-point below. A marked version of the manuscript is provided. We believe that the revised manuscript is greatly improved and meets all the requirements for publication in Materials. I hope you find our responses and revisions satisfying and thank you for your kind contributions.

Sincerely yours

Liangang Xiao, Professor
School of Materials and Energy
Guangdong University of Technology
Guangzhou, 510006, China
Email: xiaolg@gdut.edu.cn  

  • The dielectric constant values are far too precise the values in my opinion should be revised to ensure maximum one decimal place precision

Response: The Dk value of 3.4 for conventional commercial PI films in the introduction was changed to 3.49 (in precise agreement with the tested Dk value), because this experiment uses ternary polymerization and the incorporation of new monomer NADA is limited, if it is precise to one decimal, it may be misleading to the analysis of its performance change trend.

  • -The collected results should be compared to what is known from the literature

Response: Since this experiment takes the content of new monomer NADA as the variable and focuses on the influence of its structure and molar amount, its performance is more inclined to compare with the control group PI-0 and further explore a more reasonable formulation.
